# The Analysis of Superelasticity and Microstructural Evolution in NiTi Single Crystals by Molecular Dynamics

**DOI:** 10.3390/ma12010057

**Published:** 2018-12-24

**Authors:** Hung-Yuan Lu, Chih-Hsuan Chen, Nien-Ti Tsou

**Affiliations:** 1Department of Materials Science and Engineering, National Chiao Tung University, Ta Hsueh Road, Hsinchu 300, Taiwan; hungyuan0113@gmail.com; 2Department of Mechanical Engineering, National Taiwan University, Taipei 106, Taiwan; chchen23@ntu.edu.tw

**Keywords:** shape memory alloys, superelasticity, molecular dynamics, crystal variants

## Abstract

Superelasticity in shape memory alloys is an important feature for actuators and medical devices. However, the function of the devices is typically limited by mechanical bandwidth and fatigue, which are dominated by the microstructures. Thus, in order to correlate the mechanical response and the microstructures, the microstructural evolution in NiTi single crystals under the compression, tensile, and shearing tests is simulated by molecular dynamics (MD) in the current study. Then, the martensite variant identification method, which identifies the crystal variants/phases of each lattice based on the transformation matrix, is used to post-process the MD results. The results with the detailed information of variants and phases reveal many features that have good agreement with those reported in the literature, such as X-interfaces and the transitional orthorhombic phase between the austenite and monoclinic phases. A new twin structure consisting of diamond and wedge-shaped patterns is also discovered. The macroscopic behavior, such as stress-strain curves and the total energy profile, is linked with the distribution of dislocation and twin patterns. The results suggest that the loading cases of shear and compression allow a low critical strain for the onset of martensitic transformation and a better superelasticity behavior. Therefore, the two loading cases are suitable to apply to the NiTi actuators. The current work is expected to provide insight into the mechanical responses and design guideline for NiTi shape memory alloy actuators.

## 1. Introduction

Shape memory alloys (SMAs) have special mechanical properties that differ from other alloys, such as superelasticity. It is a well-known phenomenon in SMAs, enabling a high recoverable strain. Thus, SMAs have been widely applied to sensors, actuators, and biomedical devices [1,2,3]. The behavior of superelasticity and the material properties are dominated by the reorientation of the martensite phases and the evolution of the microstructures in the crystals [4]. The microstructures tend to adopt twinned structures, which minimize the energy of the crystals. Mehrabi et al. [5] also pointed out that the twinned and detwinned structures play an important role in the level of the shape recovery. However, the detail of microstructural evolution is typically difficult to observe experimentally. Thus, molecular dynamics (MD) simulations have been used to predict the material behaviors and corresponding twin patterns of the martensite phases. The results are dynamic and time-dependent, giving the opportunity to reveal the special features of SMAs. Kastner et al. [6] used MD to simulate the martensite microstructure pathways during the cyclic loading process. Pun and Mishin [7] adopted an embedded-atom interatomic potential to observe the martensitic transformation in Ni-rich NiAl alloys under a uniaxial load. Dmitriev et al. [8] studied the temperature-induced martensitic transformation in NiTi alloy and concluded that the dislocations serve as the nucleation centers for the martensite phases. This feature was also generated in the current study, that the dislocations occurred at the interfaces of martensite twins where the martensite phases initiated, having good agreement.

The results of MD simulations are largely dependent on the potential adopted by the researchers. Two types of potential have been widely used for the simulation of SMAs, i.e., the Finnis–Sinclair (FS) potential [9] and the second nearest-neighbor modified embedded-atom method (2NN MEAM) [10]. Mutter and Nielaba [11] used the FS potential to examine the austenite-martensite interfaces in NiTi nanoparticles. Sushko et al. [12] studied the deformations under different types of the indenter. However, certain phases and mechanical responses predicted by the FS potential were not in good accordance with those reported in the literature. On the other hand, the 2NN MEAM potential was modified from a many-body potential [9,13]. The potential of this type further considered the second nearest-neighbor interactions and the angular related terms [14], and thus, it may provide a more accurate stress- or temperature-induced phase transformation in NiTi alloys [10]. Thus, 2NN MEAM potential has become a well-accepted potential form for describing the NiTi binary alloys’ mechanical behaviors recently.

Many related studies examined the MD predicted microstructures by using the conventional common neighbor analysis (CNA) [15] or polyhedral template matching (PTM) [16] post-processing methods, or determining the angles of the distorted lattices. Thus, only the interfaces between the austenite and martensite phases can be clearly revealed. To understand the detailed twinning microstructure and the corresponding evolution mechanism, the crystal systems and the crystal variant pairs present in the crystals must be identified. It is well known that the variants are the results of the distortion of the lattices from the reference austenite phase toward certain directions during the martensitic transformation. Such distortion reduces the symmetry of the lattice, resulting in the distinct variants in the crystal systems, e.g., 12 variants in the monoclinic system and six variants in the orthorhombic system. Yang and Tsou [17] developed a post-processing method for MD simulations that can identify multiple crystal variants and phases by examining the transformation matrix of each lattice. Then, the detailed microstructures, twin patterns, and volume fraction of each variant of the crystals under applied boundary conditions simulated by MD can be determined.

In the current work, a standard MD simulation package (LAMMPS) and the post-processing method proposed by Yang and Tsou [17] were used to examine the superelasticity and the corresponding microstructural evolution in the equiatomic NiTi alloys, subjected to the three loading processes, i.e. tensile, compression, and shearing. Many interesting features were predicted, having good agreement with those reported in the literature. Some new microstructures were also discovered during the superelasticity process. The relationship between the microstructural evolution and the stress-strain curves of the three loading cases is discussed in the following sections.

## 2. Methodology

### 2.1. Model Setting

In the current work, stress-induced martensitic transformation and the behavior of superelasticity in the equiatomic NiTi SMAs were studied by the MD simulations. Three cases were considered here, uniaxial tensile, compression, and shearing. The simulations were performed by a well-known MD package, LAMMPS (Large-scale Atomic/Molecular Massively Parallel Simulator) [18]. The interatomic potential of NiTi was described by the 2NN MEAM potential energy [19]. The lattice constant $a_{0}$ of NiTi was set as 2.999 Å [10]; the cutoff radius $r_{c}$ was set as 5.0 Å.

The simulation box was a 18.9×18.9×18.9 nm${}^{3}$ cube. Atoms Ni and Ti were placed at a relative position at (0 0 0) and (0.5 0.5 0.5), respectively, in the austenite coordinate ([100], [010], [001]). Then, the atom pair was repeated to fill the entire simulation box, giving 62 unit cells each side and 500,094 atoms in total in the box. The periodic boundary condition (PBC) was applied in all three directions, *x*, *y*, and *z*. The atom arrangement was then relaxed at 0 K by the conjugate gradient method in order to minimize the total energy and to eliminate the internal stress. Next, the model underwent the thermal equilibrium at 325 K for 40 ps (20,000 time steps); where the velocities were assigned to atoms by the Gaussian distribution based on the given temperature. Note that the given temperature 325 K, which is between the austenite finish temperature ($A_{f}$) and the martensite start temperature ($M_{s}$), was chosen here to obtain a more stable stress-induced martensitic transformation between austenite and martensite phases in the simulation. [20].

For each case, the time step was 2 fs, and the constant temperature was controlled by the isothermal-isobaric (NPT) canonical ensemble along with a pressure of 1.013 bar. After the thermal equilibrium stage, the deformation of the material was controlled by the strain rate of the simulation box. The directions of the straining were uniaxial compression and tensile tests along the *y* direction and a shearing along the *xy* direction. The strain rate was set as $3\times 10^{8}$ s${}^{-1}$; the interval between each time step was 250 ps. In the cases of tensile and compression, the maximum strain was set as 7.5%, while it was set as 12% for the case of shearing. The box was then released gradually for another 125,000 time steps with the same rate. With the MD model built as described above, the stress-strain values during the process of the compression, tensile, and shearing were calculated.

### 2.2. Martensite Variant Identification Method

The microstructural evolution in SMAs simulated in the current study was post-processed and analyzed by the conventional visualization algorithms provided by the well-known visualization tool OVITO, such as polyhedral template matching (PTM), dislocation analysis (DXA), and atomic strain function based on the deformation gradient of lattices. These conventional tools provided the information about the phase transition, interfaces, lattice structure, and deformation. However, the limitation of those tools may not reveal the complex features of the coexistence of multi-phases and multi-variants during martensitic transformation in SMAs. Thus, the current study adopted a martensite variant identification method proposed by Yang and Tsou [17], which identifies the phases and variants based on the transformation matrix of each lattice in the single crystal. Thus, the detailed microstructure in SMAs can be revealed.

The procedures of the martensite variant identification method is illustrated in Figure 1. The method identified the phases and variants by examining each group of eight cubic unit cells (Figure 1a) in the model, which consisted with six distorted tetragonal unit cells (two for each of the *x*, *y*, *z* axes, as shown in Figure 1b–d) [21] under the given loads. The transformation matrices of the distorted tetragonal unit cells can be determined, i.e., $U_{x1},U_{x2},\cdots ,U_{z2}$, as shown in Figure 1e. Next, the similarity of the determined transformation matrices was checked (Figure 1f) with the ideal/theoretical transformation matrices of each variant in the orthorhombic (*O* phase) and monoclinic (*M* phase) crystal systems listed in the literature [22]; where the similarity was defined and quantified by the 15 conditions modified from Yang and Tsou [17] as shown in Table 1; where $u_{ij}$ is the component (i,j) of the transformation matrix U; ϕ is the tolerance with the value of 0.012, which allows more than 99% of the lattices in the crystal at a temperature above the $A_{f}$ to be identified as the austenite phase (*A* phase).

Table 2 is the characteristic table showing the conditions that are satisfied by the ideal/theoretical transformation matrix of variants in the phases *A*, *O*, and *M*. The level of the similarity of each lattice generated by MD simulations can then be computed by counting the number of matched conditions. The crystal variant of that lattice can then be identified as the variant with the most similar ideal transformation matrix (Figure 1g). The lattice is marked as “*Other*” when the similarity among all the variants is lower than 50% (Figure 1h). The algorithm described above has been implemented as a plug-in in the visualization tool OVITO. The detailed post-processing procedures can be found in the authors’ previous work [17,21].

## 3. Results and Discussion

### 3.1. The Compression Test

The results of the compression test along the *y* axis were generated. The NiTi crystal at the initial state was built in the austenite phase as described in Section 2.1. The austenite phase in the model was identified as the body-centered cubic (BCC) lattice structure by PTM, marked in blue, as shown in Figure 2a, and in white by the martensite variant identification method, as shown in Figure 2b. Initially, the whole model did not have dislocation, and the deformation gradient value was zero.

The superelasticity behavior was successfully simulated in the NiTi MD model, giving an obvious phase transition between the austenite and martensite phases. In order to verify the validity and to demonstrate the advantage of the martensite variant identification method, a herringbone twin structure, which is typically found in the experiments, during the unloading process, at about a strain of 6%, was chosen to be visualized by the PTM, deformation gradient, the martensite variant identification, and the DXA (dislocation analysis) method, as shown in Figure 3a–d, respectively. Figure 3a shows that most of the lattices had a hexagonal close-packed (HCP) structure, while some lattices were BCC or face-centered cubic (FCC), forming thin interfaces in the crystal. As reported in the literature [23], lattices of both the *O* and *M* phases had the HCP structure, and those of the *A* phase had the BCC structure. Thus, the PTM method can capture the martensitic transformation in SMAs; yet, it cannot reveal the martensite twin structure. Figure 3b shows the xy component of the deformation gradient, revealing a herringbone twin pattern within the region of HCP identified by the PTM method previously; where red and blue colors correspond to the positive and negative values of the deformation gradient. However, the number of variants presented in the herringbone pattern is still unknown. In Figure 3c, four monoclinic variants *M5*–*M8*, which form the herringbone structure, were successfully identified by the martensite variant identification method. This four-variant herringbone structure resulting from the compressive loads can also be found experimentally. The simulation results here had very good agreement with those in the literature [24]. The position and alignment of some atoms in white in Figure 3c, which were identified as the *A* phase, also had good agreement with those identified as BCC lattices. Figure 3d shows the result of the dislocation analysis. It can be found that the dislocation tended to accumulate at the twin interfaces.

Similar arrangements can be observed in Figure 3a–d, and the martensite variant identification method was able to reveal sufficient information and details in the MD results. This shows the validity and the advantage of the method. Thus, it will be used to analyze the microstructure in the following section.

Figure 4 shows the stress-strain curve and the corresponding microstructural evolution due to the compression along the *y* axis. The initial state of the crystal was in the *A* phase, as shown in Figure 4a. Then, the crystal showed a linear stress-strain relationship, and the stress dropped due to the distorted lattices, indicated in blue at Stages (b) and (c). Note that these distorted lattices were identified as variants *O3* and *O4*, further forming a stripe twin pattern, as shown in Figure 4d. Next, the two *O* phases transformed to the *M5*–*M8* variants abruptly, indicated in the red circle in Figure 4e. This phase transition also caused a slight drop in the stress. It is worth noting that the results showed that the *O* phase served as a transitional phase between the *A* phase and *M* phases, i.e., the cubic lattice of the *A* phase firstly enlarged along a direction to become the *O* phases and then switched again to one of the *M* variants. This phenomenon was also reported in the literature [23].

As the magnitude of the compressive strain increased, the crystal adopted a self-accommodated rank-two herringbone twin structure. Then, the curve underwent another linear region, and the crystal maintained the same herringbone structure. However, the maximum strain of 7.5% introduced large distortion on some lattices in Stage (f), which deviated from the *M* phase, giving some regions in the crystal that were identified as the “*Other*” phase, as shown in Figure 4f.

Stages (g)–(k) are the microstructures during the unloading procedure. The stress-strain curve again had a linear behavior, and the herringbone pattern became clear with the decreased strain. As the magnitude of strain kept decreasing, the region of the *A* phase (in white) located at the interfaces expanded, as shown in Figure 4g–i. At Stage (j), the martensite twins and the *A* phase formed the X-shaped region indicated by the red lines. This feature has good agreement with the theoretical finding of X-interfaces reported in the literature [25,26]. Finally, the martensite twins directly transformed back to the *A* phase without involving significant transitional *O* phases. The results showed that the superelasticity behavior due to the compression process was successfully performed in the current simulation.

The total energy of the entire crystal and the dislocation during the compression test are shown in Figure 5. The labels (a)–(k) in the figure correspond to those in the stress-strain curve shown in Figure 4. Each label presents a different strain level with a significant transformation. It can be observed that the presence of the *O* phase decreased the energy linearly, as shown in Figure 5b–d. At the same stage, the dislocation appeared at the interface of two *O* phases, as shown in Figure 5d. Next, around Stage (e), an abrupt energy drop occurred due to the transformation from the *O* phase to the *M* phase and the occurrence of dislocations at the twin interfaces. As almost all the regions in the crystal were in the *M* phase, the herringbone twin structure was formed, resulting in a relatively low energy state. It is worth noting that the compression did not alter the herringbone pattern; however, it greatly promoted the dislocations at the twin interfaces. The energy profile between Stages (e) and (f) shows that the occurrence of each set of dislocation can reduce the overall energy, giving a fluctuation curve in Figure 5. After the maximum strain was reached, the lowest energy herringbone twin patterns remained until the X-interfaces introduced by the development of the *A* phase. Finally, the crystal returned to the state with no dislocation and with exactly the same energy value as its initial state.

### 3.2. The Tensile Test

The stress-strain curve and the corresponding microstructures of the uniaxial tensile test along the *y* direction are shown in Figure 6. The crystal adopted a perfect austenite structure initially and then underwent a linear elongation. Then, the stress decreased slightly when the strain reached about 3.0%. This is because of the phase transformation from the *A* phase to the *M* phase. It is worth noting that there was no significant *O* phase involved in the martensitic transformation in the tensile case compared with the case of the compression test. Thus, there was no obvious plateau in the curve in this case. The crystal then quickly adopted the herringbone pattern consisting of variants *M1*–*M4*. It is interesting that, when the strain was about 5.0%, the crystal adopted another self-accommodating microstructure, which led to a significant stress reduction, as shown in Figure 6d. Based on the martensite variant identification method, this microstructure can be regarded as two diamond-shaped martensite regions (*M1*–*M2* and *M3*–*M4* twins) surrounded by wedge-shaped regions (identified as the *A* phase). This interesting microstructure will be discussed in more detail later. When the maximum tensile strain 7.5% was reached, many lattices in the crystal were largely distorted, and thus identified as the “*Other*” phase. Even so, the microstructure remained unchanged (Figure 6f,g) until the strain returned to about 3.5%.

At Stage (h), the crystal returned to the herringbone structure, which was similar to that at Stage (c). Thus, the unloading curve declined linearly with the same rate as the loading curve inclined around Stage (c). Next, the band-shaped region of the *A* phase, marked in white in Figure 6i,j, expanded and further transformed into wedge shapes, as shown in Figure 6k. As the unloading procedure finished, the crystal did not fully return to the *A* phase, but adopted a microstructure that was similar to that at Stage (g), which consisted of both martensite twins and the *A* phase. This indicates that the superelasticity induced by the tensile loading was less significant than that induced by the compression loading.

The microstructure induced by the tensile strain around 6% was not the typical twin pattern reported in the literature. The cross-section views generated by PTM, DXA, and the martensite variant identification methods are shown in Figure 7. In Figure 7a, the PTM method identified the wedge-shaped region as FCC, with a small portion of BCC lattices at the boundaries. Those boundaries of the wedges also accumulated dislocations, as shown in Figure 7b. The results generated by the martensite variant identification are shown in Figure 7c. It can be observed that the regions of HCP identified by PTM can be categorized into two types of twin structure, *M1*–*M2* twins and *M3*–*M4* twins, while those wedge regions were identified as the *A* phase. However, based on the literature [23], a typical lattice of the *A* phase has the BCC structure. This indicates that the transformation matrix of the lattices within the wedge regions was very similar to that of the ideal/theoretical austenite phase, and yet, the lattices were still distorted toward FCC from BCC. Thus, it is believed that these lattices within the wedge regions were in the highly stressed austenite phase, and this may be the cause of the residual stress after the removal of the loads.

The total energy and the dislocation during the tensile test are shown in Figure 8. It can be observed that the tensile loads induced a direct transformation from the austenite to monoclinic phases. The occurrence of the martensite twins significantly minimized the total energy, and dislocation appeared at the interface of the twin, as shown in Figure 8b. Similar to the case of the compression test, the energy profile between Stages (b) and (c) was fluctuating due to the formation of the dislocations between the herringbone twin interfaces. At Stage (d), the arrangement of the dislocations changed significantly, which stabilized the energy curve and allowed a linear incline of the curve as the loading continued, as shown in Figure 8e–f. Then, the arrangement of the dislocations changed again, accumulating at the herringbone twin interfaces (Figure 8g–j). Finally, as the strain returned to zero, the dislocations remained, and only parts of the crystal returned to the *A* phase.

### 3.3. The Shearing Test

Figure 9 shows the stress-strain curve and the corresponding microstructural evolution in the case of the shearing test. The crystal in the *A* phase (Figure 9a) was subjected to the *xy* shear straining. Similar to the compression test, the *A* phase firstly transformed into the *O* phase (variants *O5* and *O6*), as shown in Figure 9b, and then further transformed into the *M* phase (variants *M9* and *M12*), as shown in Figure 9c. The four variants formed a typical herringbone twin structure. It can be observed that the applied shearing favored *M11* and *M12* only and that the region of *M9*–*M10* twins was consumed by the *M11*–*M12* twins, as shown in Figure 9d–e. The resulting structure was a rank-one twin pattern, i.e., alternating strips of variants. When the shear strain reached the maximum value of 12%, the microstructure remained in the form of rank-one twins, yet with some highly disordered lattices, as shown in Figure 9f.

Stages (g)–(k) were the unloading procedure. At Stage (h), some regions returned to the *A* phase, and variant *O6* served as the transitional phase between the *A* phase and *M11* and *M12*. Then, the *A* phase kept expanding by consuming the *M11*–*M12* twins, as shown in (i)–(j). At Stage (k), most parts of the crystal were in the *A* phase and variant *O6*. Finally, the remaining variant *O6* vanished, and the crystal returned to the *A* phase as the initial state. This also indicates that the superelasticity induced by the shearing test was successful.

The energy profile and the distribution of the dislocations are shown in Figure 10. It can be observed that the onset of the martensitic transformation due to the shearing (around 1%) was earlier than the compression and tensile test. When the *O* phase appeared (Figure 9b), a small amount of dislocations occurred, as shown in Figure 10b. Then, the dislocations significantly increased at the twin interfaces in the herringbone structure as the crystal transformed from the *O* phases to the *M* phases, as shown in Figure 10c–d. At the same stage, the total energy also decreased abruptly. It is interesting that although the crystal was stressed by the further shearing, the energy was minimized due to the change of the type of dislocations, as shown in Figure 10e. After Stage (e), the dislocations migrated toward each other when shear strain increased and moved away from each other during the unloading, as described in literature [8]. This enabled a steady and linear increase of the energy, and the crystal returned to the original state without any residual dislocation (Stages (g)–(k)).

### 3.4. Discussion

The stress-strain response, microstructural evolution, energy profile, and dislocation distribution for the compression, tensile, and shearing tests were reported. They revealed several features that have good agreement with the experimental results. In the current study, the dislocations were mainly observed at the interfaces between the austenite and martensite phases. This feature corresponds to the compatibility between the two phases/twins at the interface, as described by the cofactor conditions [27]. The cofactor conditions predicted that the compatibility between the austenite and martensite becomes better if the middle eigenvalue of the transformation stretch matrices, $\lambda_{2}$, is close to one. Under this condition, the stress field at the *A*/*M* interface can be minimized and thus prevents the formation of dislocation during phase transformation. An experimental study has shown that the $\lambda_{2}$ value of NiTi is about 0.9679, which deviates from one apparently [28]. As a consequence, the stress field at the *A*/*M* interface of NiTi was considered relatively large, and thus, dislocations forming at the interface can be observed in the model. In addition, experimental results also support that dislocations were introduced into NiTi during the phase transformation, which will result in a reduction in transformation temperatures [29,30] or a decrease in the stress to induce martensitic transformation [31,32].

As for the onsets of the martensitic transformation, an experimental study on polycrystal NiTi showed that the critical stress to induce martensitic transformation from the austenite phase was the lowest for shear deformation, followed by compression and then tension [33]. This was in good accordance with the predicted onset of martensitic transformation in the cases of shearing, compression, and tensile simulations (about at a strain of 1%, 1.5%, 2.7%, respectively). This feature can also be explained by the dislocation density around the transition point. Figure 5, Figure 8 and Figure 10b showed that the dislocation density was the lowest under the shear deformation, while it was the greatest in the tensile test.

## 4. Conclusions

The uniaxial compression, tensile, and shearing tests for NiTi alloys were studied by MD simulations in the current work. With the help of the martensite variant identification method, the simulation results revealed several features, having good agreement with the experimental and theoretical results reported in the literature, such as X-interfaces during the martensitic transformation, herringbone twin patterns, and most importantly, the orthorhombic phase serving as the transitional phase between the austenite and the monoclinic phases.

The relationship between the interface compatibility in NiTi and the dislocation accumulation was also discussed. The tendency of the onset of martensitic transformation for the shearing, compression, and tensile tests (from low to high) was also accurately predicted. Our simulations also showed that the superelasticity of the NiTi single crystal was significant for the shear and compression loading and was not obvious for the tensile case. By using three post-processing methods, PTM, DXA, and the martensite variant identification method, the microstructures with wedge-shaped regions of the stressed austenite phase were revealed. The special microstructure accumulated dislocations on the twin boundaries and is expected to be the cause of the reduction of the superelasticity of NiTi under the tensile loads. The relationship between energy profile and dislocation distribution was also discussed. The current work systematically studied the superelasticity and the corresponding microstructural evolution in a NiTi single crystal by MD simulation. The results presented here can provide design guidelines for SMA applications; where the superelasticity can be enhanced by applying proper shear or compressive loads; the occurrence of the irreversible dislocation, which may reduce the life cycle of the devices, can be avoided. 

## Figures and Tables

**Figure 1 materials-12-00057-f001:**
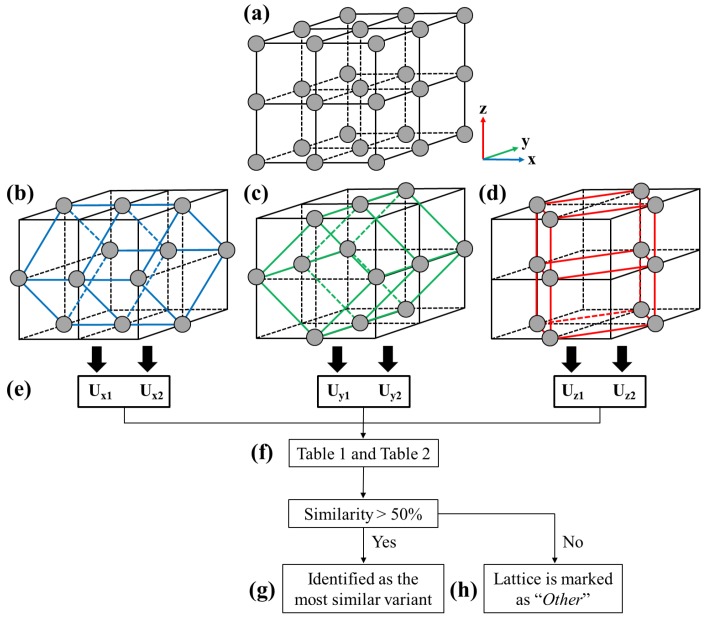
The procedures of the martensite variant identification method. (**a**) the eight cubic unit cells, (**b**–**d**) the six tetragonal unit cells, (**e**) the transformation matrices of the six tetragonal unit cells, (**f**) conditions checked in Table 1 and Table 2, and (**g**,**h**) the variant can be identified.

**Figure 2 materials-12-00057-f002:**
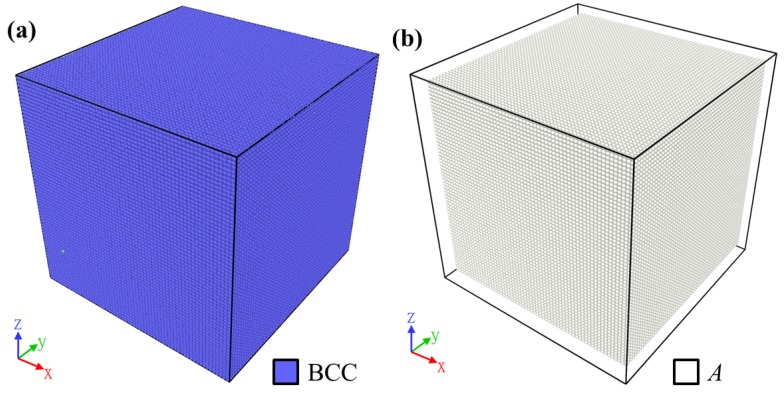
A typical example of the results at the initial step of the compression tests, analyzed and visualized with (**a**) polyhedral template matching (PTM) and (**b**) the martensite variant identification.

**Figure 3 materials-12-00057-f003:**
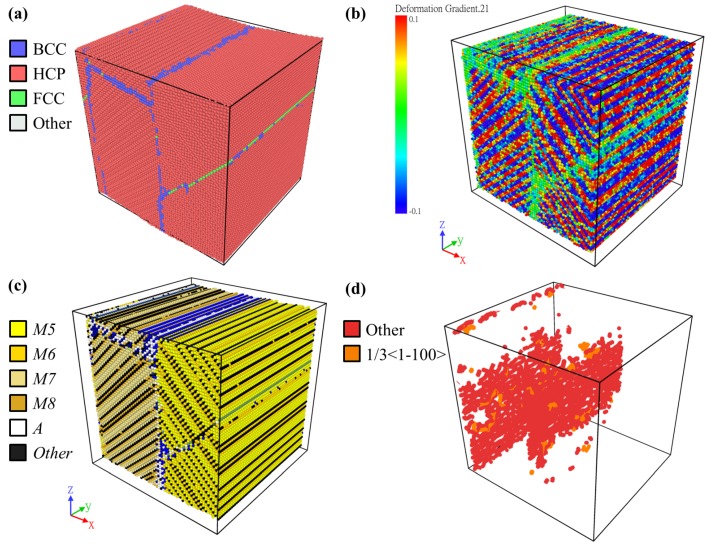
A typical example of the results during the compression tests, analyzed and visualized with (**a**) PTM, (**b**) the deformation gradient, (**c**) the martensite variant identification, and (**d**) DXA (dislocation analysis) methods.

**Figure 4 materials-12-00057-f004:**
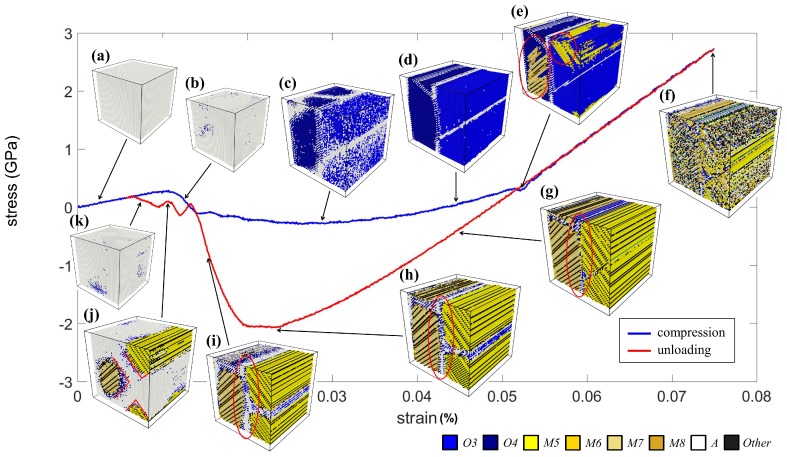
The stress–strain curve of the NiTi shape memory alloy (SMA) subjected to the compression process. The corresponding microstructures at stages (**a**–**k**) are shown.

**Figure 5 materials-12-00057-f005:**
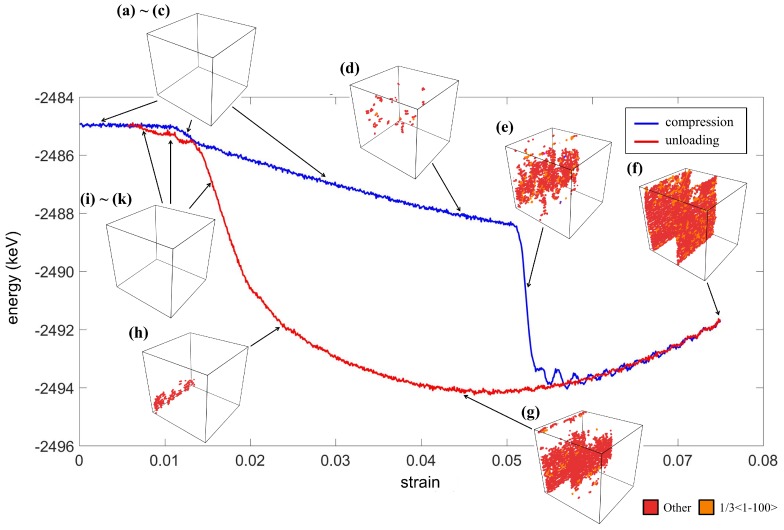
The energy profile and the evolution of dislocation at stages (**a**–**k**) during the compressive loading.

**Figure 6 materials-12-00057-f006:**
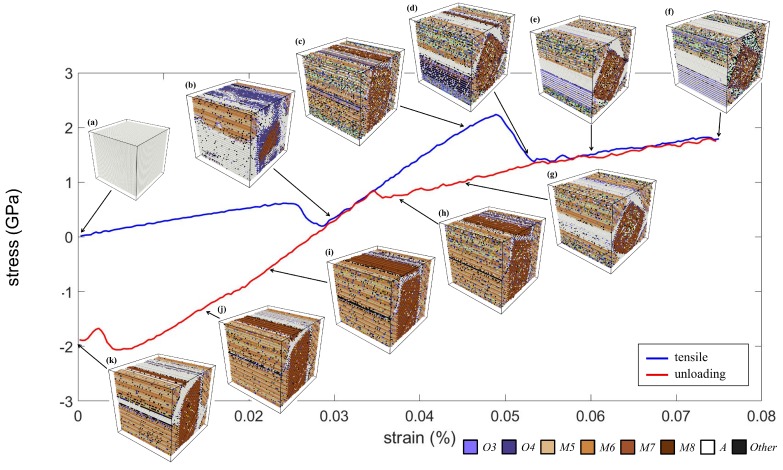
The stress–strain curve of the NiTi SMA subjected to tensile loading. The corresponding microstructures at Stages (**a**–**k**) are shown.

**Figure 7 materials-12-00057-f007:**
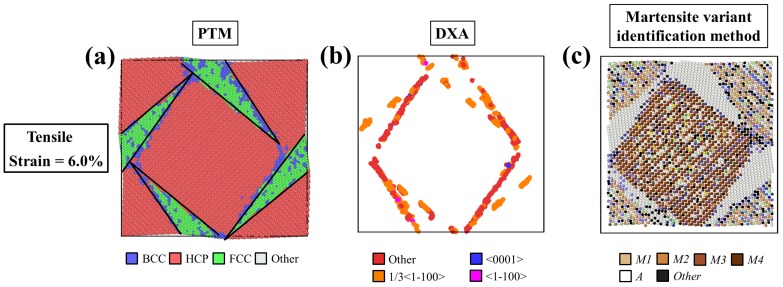
The self-accommodating microstructure induced at the tensile strain of 6%, visualized by (**a**) PTM, (**b**) DXA (dislocation analysis), and (**c**) the martensite variant identification methods.

**Figure 8 materials-12-00057-f008:**
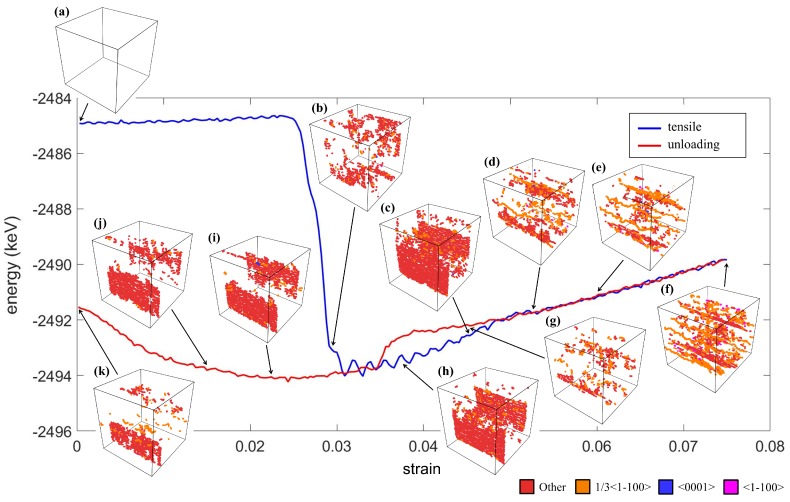
The energy profile and the evolution of dislocation at Stages (**a**–**k**) during the tensile loading along the *y* axis.

**Figure 9 materials-12-00057-f009:**
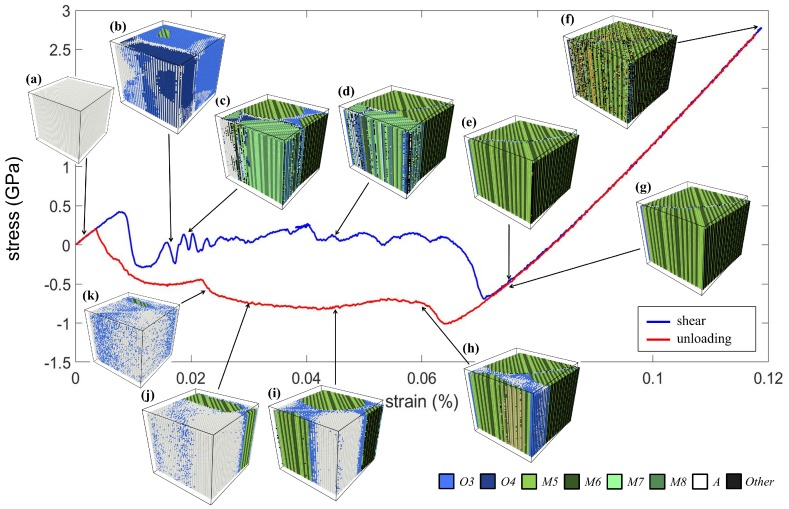
The stress-strain curve of the NiTi SMA subjected to the *xy* shearing load. The corresponding microstructures at Stages (**a**–**k**) are shown.

**Figure 10 materials-12-00057-f010:**
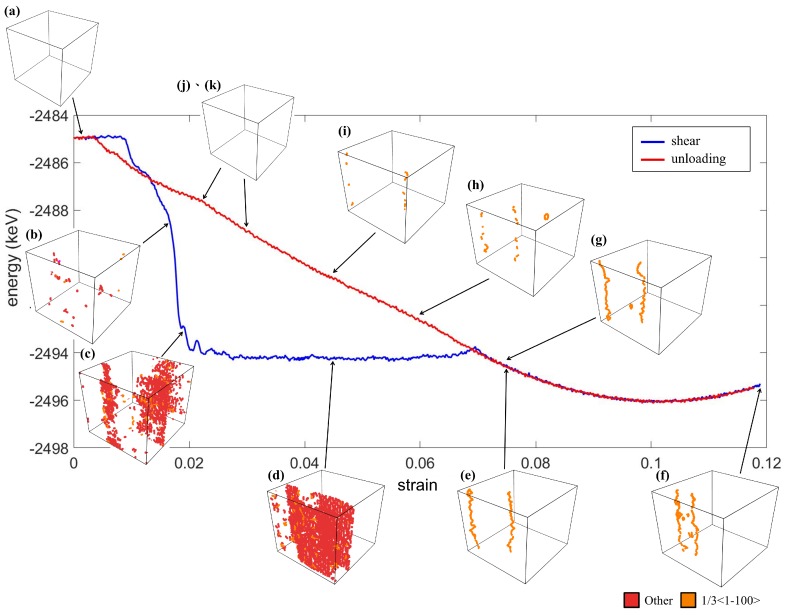
The energy profile and the evolution of dislocation at Stages (**a**–**k**) during the *xy* shearing.

**Table 1 materials-12-00057-t001:** The 15 conditions for identifying variants.

Condition ID	Conditions
A	$u_{11}-1\geq \phi$
B	$u_{22}-1\geq \phi$
C	$u_{33}-1\geq \phi$
D	$u_{11}-1<\phi$
E	$u_{22}-1<\phi$
F	$u_{33}-1<\phi$
G	$u_{12}\geq \phi$
H	$u_{13}\geq \phi$
I	$u_{23}\geq \phi$
J	$u_{12}\leq -\phi$
K	$u_{13}\leq -\phi$
L	$u_{23}\leq -\phi$
M	$\left|\right|u_{12}\left|-\right|u_{13}\left|\right|\leq \phi$
N	$\left|\right|u_{12}\left|-\right|u_{23}\left|\right|\leq \phi$
O	$\left|\right|u_{13}\left|-\right|u_{23}\left|\right|\leq \phi$

**Table 2 materials-12-00057-t002:** The satisfied conditions for the austenite (*A*) phase and each variant of the orthorhombic (*O*) and monoclinic (*M*) phases.

Variant No.	Conditions to be Satisfied
*M1*	B, C, D, I, J, K, M
*M2*	B, C, D, G, H, I, M
*M3*	B, C, D, G, K, L, M
*M4*	B, C, D, H, J, L, M
*M5*	A, C, E, H, J, L, N
*M6*	A, C, E, G, H, I, N
*M7*	A, C, E, G, K, L, N
*M8*	A, C, E, I, J, K, N
*M9*	A, B, F, G, K, L, O
*M10*	A, B, F, G, H, I, O
*M11*	A, B, F, I, J, K, O
*M12*	A, B, F, H, J, L, O
*O1*	B, C, D, I
*O2*	B, C, D, L
*O3*	A, C, E, H
*O4*	A, C, E, K
*O5*	A, B, F, G
*O6*	A, B, F, J
*A*	M, N, O

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
