# Peer review of "The Analysis of Superelasticity and Microstructural Evolution in NiTi Single Crystals by Molecular Dynamics"

_materials, 2018, doi:10.3390/ma12010057_

Reviewer 1 Report

This manuscript describes a microstructural evolution in NiTi single crystals under the compression, tensile and sharing test. This behaviour has been simulated by molecular dynamics. The reported results reveal features that corresponds with the one reported in literature, i.e. X-interfaces and the transitional orthorhombic phase between the austenite and monoclinic phase.

Overall, the manuscript is well written and described. It is easy to read and, as suggested by the authors, it can be useful for the design of SMA actuators.

However, there are still same sections that can be improved as reported in the following points.

Point 1

Abstract

Page 1, Line 1

The abstract should introduce the context of the manuscript with more information. It would be better to introduce the advantages of SMA for the design of an actuator and describe the limitation in the state of the art (e.g. mechanical bandwidth, etc.). Only a few lines will be enough.

Page 1, Line 5.

The acronym MD should be defined.

Point 2

Page 1, Line 17

Additional references should be included for SMA sensors, actuators and biomedical devices. Following some references to be included:

Sensors:

[1] Wang, T.-M.; Shi, Z.-Y.; Liu, D.; Ma, C.; Zhang, Z.-H. An Accurately Controlled Antagonistic Shape Memory Alloy Actuator with Self-Sensing. Sensors 2012, 12, 7682-7700.

Actuators:

[2] Manfredi, L.; Huan, Y, Cuschieri, Low power consumption mini rotary actuator with SMA wires. Materials , Smart Materials and Structures, Sep 2017

Biomedical Devices/Review:

[3] Khoo, Z.X.; Liu, Y.; An, J.; Chua, C.K.; Shen, Y.F.; Kuo, C.N. A Review of Selective Laser Melted NiTi Shape Memory Alloy. Materials 2018, 11, 519.

Point 3

Several acronyms are missing. Following some examples.

Page 3, line 80, NPT

Page 8, line 209, HCP, PTM

Page 8, line 211, BCC

Point 4

Page 3, line 116.

The algorithm described in section 2.2 could be included in the manuscript. Perhaps in a diagram in the section and also, if not too long, the main code in an additional appendix at the end of the manuscript, or in a public repository. This will help a reader to replicate the simulation results.

Point 5

Page 5, line 157

Stage “f” is missed in the description of section 3.1. Please include it.

Point 6

In figures 2, 3, 5, 7, 9, 10 the title of X axis reports “%”, but in the graphs the reported value are not, i.e. 0.08 should be 8.

Point 7

Page 13, line 296

In the conclusion, the authors report a sentence which is exactly reported in the abstract” “The results are expected to provide insight to the mechanical responses and design guideline for SMA actuators.” Please, rephrase it.

In addition, the authors should give more information about this statement, such as advices on how this manuscript contribution could be used as a guideline for SMA actuators design.

Author Response

Response to Reviewer 1 Comments

Point 1:

Abstract: Page 1, Line 1: The abstract should introduce the context of the manuscript with more information. It would be better to introduce the advantages of SMA for the design of an actuator and describe the limitation in the state of the art (e.g. mechanical bandwidth, etc.). Only a few lines will be enough.

Page 1, Line 5.: The acronym MD should be defined.

Response 1: The description about the limitation, such as mechanical bandwidth and fatigues, related to the design of SMA actuators is added and marked in red in the abstract. We also added that, based on the results, the two loading cases (shear and compression) are recommended to apply to the NiTi actuators. Also, the full name of MD is added in Page 1, Line 5.

Point 2: Page 1, Line 17: Additional references should be included for SMA sensors, actuators and biomedical devices.

Response 2: Thanks for the reviewer’s helpful input. The references for sensors, actuators and biomedical device are added in the first paragraph in section 1. Page 1, Line 21.

Point 3: Several acronyms are missing. Following some examples.

Page 3, line 80, NPT

Page 8, line 209, HCP, PTM

Page 8, line 211, BCC

Response 3: NPT, HCP, PTM, BCC are the acronyms of isothermal-isobaric ensemble, hexagonal close-packed, polyhedral template matching and body-centered cubic, respectively. The full names are added in the texts.

Point 4: Page 3, line 116: The algorithm described in section 2.2 could be included in the manuscript. Perhaps in a diagram in the section and also, if not too long, the main code in an additional appendix at the end of the manuscript, or in a public repository. This will help a reader to replicate the simulation results.

Response 4: The flowchart diagram (Figure 1) which describes the algorithm of martensite variant identification method and the corresponding explanation are added in the manuscript. However, as the method has been published elsewhere (Yang and Tsou 2016), we have asked the readers to refer to the reference for more details in Page 4, Line 124.

Point 5: Page 5, line 157: Stage “f” is missed in the description of section 3.1. Please include it.

Response 5: Thanks for the suggestions from the reviewer. The description about stage (f) is added in the paragraph in Page 6, Line 170.

Point 6: In figures 2, 3, 5, 7, 9, 10 the title of X axis reports “%”, but in the graphs the reported value are not, i.e. 0.08 should be 8.

Response 6: Thanks for the suggestions from the reviewer. The “%”s in the title of X axis are removed for all figures.

Point 7: Page 13, line 296: In the conclusion, the authors report a sentence which is exactly reported in the abstract” “The results are expected to provide insight to the mechanical responses and design guideline for SMA actuators.” Please, rephrase it.

In addition, the authors should give more information about this statement, such as advices on how this manuscript contribution could be used as a guideline for SMA actuators design.

Response 7: In Page 13, Line 312, the statement of the ending of conclusion is rephrased.

We thank for the helpful suggestions from the reviewer.

Reviewer 2 Report

This is a well-written manuscript on an interesting topic, and I am happy to recommend publication in Materials

Author Response

Response to Reviewer 2 Comments

The authors thank the positive comments from the reviewer. We believe, with your and the other reviewers’ comments, the manuscript is more suitable for the publication of the current journal at the current version. Many thanks.

Reviewer 3 Report

This manuscript entitled “The analysis of Superplasticity and Microstructural Evolution in NiTi Single Crystal by Molecular Dynamics” investigates the microstructural evolution in Ni-Ti shape memory alloy under the action of external lodaing. The manuscript is well written and organized. Presented results reveal that better superplasticity behavior in NiTi can be achieved at shear and compression loading. A new twin structure is discovered. Nevertheless, there are some questionable points:

1.Authors need to enhance introduction part. Most of the Refs. are quite old which means that the subject is not interesting now or was totally studied decades ago or the applied method is not so good to study the processes.

2.The application of MD to the simulation of superplasticity behavior should be additionally proven. Though authors show that the results have good agreement with the results reported in the literature, the overall description of the subject is poor. Again, refs. On MD simulation of martensitic transformation are old, however, there are some new works on this subject (Dmitriev et al. Letters on Materials 7(4) 2017), where even more simple model is used, which are not mentioned in the manuscript. 

3.Moreover, the best way to prove the simulation results is to compare with the experiment, as it was done in (Ref. [3] from the present manuscript). The comparison of the microstructure obtained in experiment with the simulation results can additionally proof the application of this model. 

4.For Fig. 1 authors wrote that the microstructure is typical. In my opinion, at first initial structures should be presented. An some special characteristics should be also given for typical twin structure like time, strain, stress or something other. Maybe even atomic arrengement of the phases should be presented for clearance.

5.Why not to organize Figs. 3 and 4 the same as Fig. 2? If Figs. 2 and 3 for the same process than there should be strong correlation between changes in stress and energy. May be it will be better to combine these figures to show that they are correlate at the same strain values. And why not to show structure at exactly the same strain values for both curves? Authors wrote that the labels in Figs. 3 and 4 is the same as for Fig. 2, but, for example, (h) and (b) are taken at different strain or it seems like this from this presentation. The same for other loading types. Even the scale (if printed) of figures is different and it is very hard to compare and analyze it.

6.In Fig. 11 empty boxes look strange. I can understand why they are shown but in my opinion it is better just to mention that they are empty. In Fig. 10 curves starts from 0, but strain is shown from -0.2. It seems that it is because of Fig. 9, and because of the structures presents in Fig. 9, but it have no physical meaning.

Nevertheless, present manuscript can be accepted after revision.

Author Response

Response to Reviewer 3 Comments

Point 1: Authors need to enhance introduction part. Most of the Refs. are quite old which means that the subject is not interesting now or was totally studied decades ago or the applied method is not so good to study the processes. 

Response 1: Thanks for the reviewer’s useful suggestion. The references have been updated by the recent references which study similar topics to those of the old references. They are 

[4] Gao, W.;  Yi, X.;  Sun, B.;  Meng, X.;  Cai, W.;  Zhao, L.   Microstructural evolution of martensite during deformation in Zr50Cu50 shape memory alloy. Acta Materialia 2017,132, 405–415.

[5] Mehrabi,  R.;  Kadkhodaei,  M.;  Elahinia,  M.   Constitutive  modeling  of  tension-torsion  coupling  and tension-compression asymmetry in  NiTi shape memory alloys. Smart Materials and Structures 2014, 23, 075021.

[10] Ko, W.S.; Grabowski, B.; Neugebauer, J. Development and application of a Ni-Ti interatomic potential with high predictive accuracy of the martensitic phase transition. Physical Review B2015,92, 134107.

[13] Ren, G.; Sehitoglu, H. Interatomic potential for the NiTi alloy and its application. Computational Materials Science 2016,123, 19–25.

In addition, we also added more new references about the application of NiTi actuators as the reference 1 to 3.

Point 2: The application of MD to the simulation of superplasticity behavior should be additionally proven. Though authors show that the results have good agreement with the results reported in the literature, the overall description of the subject is poor. Again, refs. On MD simulation of martensitic transformation are old, however, there are some new works on this subject (Dmitriev et al. Letters on Materials 7(4) 2017), where even more simple model is used, which are not mentioned in the manuscript.

Response 2: The reference has added in the introduction, and the correlation between the reference and the current work is addressed in Page 2 Line 32.

Point 3: Moreover, the best way to prove the simulation results is to compare with the experiment, as it was done in (Ref. [3] from the present manuscript). The comparison of the microstructure obtained in experiment with the simulation results can additionally proof the application of this model.

Response 3: We have added the experimental result about NiTi (reference [24]) to validate the results generated by our simulation. The corresponding description is added in Page 5, Line 148.

Point 4: For Fig. 1 authors wrote that the microstructure is typical. In my opinion, at first initial structures should be presented. And some special characteristics should be also given for typical twin structure like time, strain, stress or something other. Maybe even atomic arrengement of the phases should be presented for clearance.

Response 4: The initial structure for the compression test is added as Fig.2. The initial structure is B2 structure and the description is also added in Page 5, Line 128. The features, such as the strain state and the four-variant composition, of the typical herringbone twin structure are described in Page 5, Line 135.

Point 5: Why not to organize Figs. 3 and 4 the same as Fig. 2? If Figs. 2 and 3 for the same process then there should be strong correlation between changes in stress and energy. May be it will be better to combine these figures to show that they are correlate at the same strain values. And why not to show structure at exactly the same strain values for both curves? Authors wrote that the labels in Figs. 3 and 4 is the same as for Fig. 2, but, for example, (h) and (b) are taken at different strain or it seems like this from this presentation. The same for other loading types. Even the scale (if printed) of figures is different and it is very hard to compare and analyze it.

Response 5: Thanks for the reviewer’s useful comments. All the energy figures and dislocation figures have been combined together as Fig.5, 8 and 10. The strain values for each label as (a) to (k) are revised. Similar problem for the figures in tension and shear cases are also solved. The scale problem about each energy and stress-strain figure are updated. 

Point 6: In Fig. 11 empty boxes look strange. I can understand why they are shown but in my opinion it is better just to mention that they are empty. In Fig. 10 curves starts from 0, but strain is shown from -0.2. It seems that it is because of Fig. 9, and because of the structures presents in Fig. 9, but it have no physical meaning.

Response 6: We combined the energy profile with the dislocation distribution as mentioned in the previous comment. Therefore, we keep only one empty box for illustration purpose to represent the dislocation-free state in stage (j) to (k). The problem about the x axis in the figures for shearing test, which start from -0.2, is fixed. The figures have been rearranged and the negative meaningless part has been removed now.

We thank for the helpful suggestions from the reviewer.